

# I feel addicted to watching TV series: association between binge-watching and mental health

Francesca Favieri[*], Giuseppe Forte[*], Renata Tambelli, Manuela Tomai and Maria Casagrande

Department of Dynamic, Clinical Psychology and Health, "Sapienza" University of Rome, Rome, Italy
[*] These authors contributed equally to this work.

## ABSTRACT

**Background**. Binge-watching (BW) is the consecutive viewing of three or more episodes of the same series in one sitting. Although some negative effects on mental health were evidenced, the continuum of BW from leisure activity to problematic behavior is still unclear. This study aimed to analyze mental health (depression, trait anxiety, social anxiety, impulsivity, alexithymia) of people involved in different expressions of BW.

**Methods**. A cross-sectional survey collected data from 482 respondents. According to a validated BW questionnaire, participants were divided into Problematic BW, Moderate BW, Non-BW, and No-viewer, and differences between groups were tested on psychological dimensions assessed *via* standardized questionnaires evaluating: trait and social anxiety, depression, impulsivity, and emotional dysregulation.

**Results**. An association between problematic BW and worse mental health conditions was evidenced, and a positive effect of non-problematic BW was supported. A negative linear trend from the BW as a leisure activity to problematic BW was marked, indicating how a possible maladaptive behavior orientation of BW in specific mental health conditions could be figured out as a behavioral addiction.

## INTRODUCTION

The way to approach the vision of TV content has drastically changed in the last few years. With the large diffusion of new technological devices and on-demand services for entertainment, viewers can choose the time and place of the fruition of TV series which are permanently available (*Granow, Reinecke & Ziegele, 2018*). Consequently, an intense and consecutive vision of several episodes of one TV series has become a popular TV-viewing pattern (*Granow, Reinecke & Ziegele, 2018*; *Snyder & Merritt, 2016*; *Steins-Loeber et al., 2020*), referred to as binge-watching (BW; *Forte et al., 2021*; *Horvath et al., 2017*; *Shim & Kim, 2018*).

The interest in BW is recently increased due to its possible negative consequences on well-being (*Flayelle, Maurage & Billieux, 2017*; *Forte et al., 2021*). Neglect of important

Corresponding authors
Francesca Favieri,
francesca.favieri@uniroma1.it
Maria Casagrande,
maria.casagrande@uniroma1.it

tasks and duties, sleeping problems, fatigue, reduced social contacts, and long-term health issues related to inactivity and unhealthy eating were recently associated with BW, as well as increasing in sedentary lifestyles and a reduction in general well-being (*Camart, Zebdi & Bouvet, 2018*; *Exelmans & Van den Bulck, 2017*). Accordingly, the common aspects between BW and other behavioral addictions (*e.g.*, pathological gambling, internet addiction, compulsive buying disorder), such as loss of control, craving, avoidance, neglect of other activities, maladaptive coping strategies, and negative emotions, were advanced (*Forte et al., 2021*; *Steins-Loeber et al., 2020*).

However, some authors underlined the double valence of BW as a phenomenon that can both negatively and positively affect individuals (*Granow, Reinecke & Ziegele, 2018*). In fact, the vision of TV series, driven by entertainment motivation (*Billieux et al., 2015*), may represent a leisure activity increasing viewers' enjoyment, perceived autonomy, personal satisfaction, and psychological well-being (*Granow, Reinecke & Ziegele, 2018*; *Troles, 2019*).

To prevent over-pathologizing of BW, highlighting the differences and similarities with addictive behavior appears important (*Flayelle et al., 2019*; *Forte et al., 2021*). Furthermore, it can be relevant to understand the transition from BW as a positive engagement to excessive and uncontrolled behavior associated with negative consequences on mental health, distress, and functional impairment in everyday life.

Similar to other behavioral addictions, some personality characteristics (*e.g.*, emotional dysregulation, loss of self-control, impulsivity, anxiety, depression, perceptual and attentional problems) might make individuals sensitive to developing pathological BW. For example, previous research evidenced an association between BW, depression, and impulsivity (*Ahmed, 2017*; *Steins-Loeber et al., 2020*; *Sun & Chang, 2021*). A recent study by *Sun & Chang (2021)* reported a positive association between problematic BW and depression, social anxiety, and loneliness. Also, confirming previous literature on TV addiction, problematic BW was associated with sleep disorders, mood decrease, and self-perception of low quality of life (*Forte et al., 2023*; *Alfonsi et al., 2023*). However, the association between psychological health, particularly considering trait personological variables, and BW is still unclear due to the differences in the definition and measurement of binge-watching in research (*Forte et al., 2021*) and the few empirical studies on this topic.

In behavioral addictions' studies, an interesting model is the Interaction of Person-Affect-Cognition-Execution (I-PACE; *Brand, Laier & Young, 2014*; *Brand et al., 2019*). This model considers predisposing variables representing core characteristics of the person, affective and cognitive responses to external or internal stimuli. This theoretical framework hypothesizes that the development of problematic behaviors (*i.e.*, addictive behaviors) is a consequence of the interactions among neurobiological aspects, predisposing psychological characteristics (*e.g.*, personality traits such as impulsivity, psychopathologies like anxiety and depression), and moderating variables such as affective and cognitive dimensions (*e.g.*, reward expectancies, coping style, implicit cognition, executive functions, and decision making). Accordingly, to assess one of the dimensions of the model—*i.e.*, the psychological one—in association with a possible new addiction as the BW is relevant for two reasons. On the one side, extending and confirming the I-PACE for the BW would

support its feature as addictive behavior. On the other side, I-PACE might furnish the frame on which an adaptive and leisure activity can develop into a behavioral addiction.

According to these premises, the current study aimed to define the mental health phenotype of problematic, moderately problematic, and non-problematic binge-watching behaviors. Accordingly, different psychological characteristics (*i.e.,* depression, trait anxiety, social phobia, social anxiety, impulsivity, and alexithymia, as emotional dysregulation index) were examined in individuals who approached the vision of TV series differently (*i.e.,* non-problematic, moderately problematic, problematic watchers) and in individuals who do not watch TV-Series. We expected the more problematic BW to be associated with higher mental distress than watching TV series as a leisure activity. Moreover, with the aim of start to move into a deeper understanding of the I-PACE model in the context of new behavioral addictions, this study considered the possible predictive association between the psychological dimensions assessed in the study toward the BW pattern.

## MATERIALS & METHODS

### Participants

Four-hundred eighty-two young adults participated voluntarily in the study (age range: 18–35; mean age = 21.63; dev.st = 3.20; 320 females and 153 males), completing an online survey. According to self-reported TV series viewing and the results of the Binge-Watching Addiction Questionnaire, the respondents were classified into four groups: (1) non-TV series viewers (No-Viewers: 78); (2) non-problematic binge-watchers (No-BW = 180); (3) moderately problematic binge-watchers (Moderate BW = 180); (4) problematic binge-watchers (Problematic BW = 44).

The main characteristics of the groups are shown in Table 1.

### Questionnaires
#### *Demographic and lifestyles questions*
A short questionnaire collected general demographic information from each respondent (*e.g.,* gender, age, educational levels, occupational and relational status).

#### *Binge-watching behavior*
The Binge-Watching Addiction Questionnaire (BWAQ; *Forte et al., 2021*) was adopted to assess BW behavior. The BWAQ comprises 24 items on 5-point Likert scales (*i.e.,* from 0 = never to 4 = always). BWAQ considers different components of the addictive nature of BW (*i.e.,* craving, dependency, anticipation, and avoidance). Moreover, it provides a global score. To define moderate or problematic BW behavior, BWAQ reports different cut-offs. A score higher than 69 indicates problematic behavior, a score equal to or higher than 51 and lower than 69 indicates moderate BW, while scores lower than 51 are associated with non-problematic BW (*Forte et al., 2021*).

#### *Anxiety*
The State-Trait Anxiety Inventory (STAI-Y; *Pedrabissi & Santinello, 1989*) was adopted to measure anxiety. STAI-Y is a 40-items questionnaire structured on 4-point Likert scales

**Table 1  Sociodemographic information of the groups.**

|  | No-Viewers | NO-BW | Moderate BW | Problematic BW |
|---|---|---|---|---|
| *Gender* | | | | |
| Male | 33 (21.6) | 74 (48.4) | 38 (24.8) | 8 (5.2) |
| Female (%) | 45 (13.7) | 106 (32.2) | 142 (43.2) | 36 (10.9) |
| *Educational Level* | | | | |
| High School | 41 (11.8) | 104 (30.1) | 160 (46.3) | 41 (11.8) |
| Bachelors Degree | 24 (22.0) | 72 (66.1) | 12 (11.0) | 1 (0.9) |
| Master Degree | 13 (48.1) | 4 (14.8) | 8 (29.6) | 2 (7.5) |
| *Relationship Status* | | | | |
| In a relationship | 38 (23.9) | 75 (47.2) | 37 (23.3) | 9 (5.6) |
| Single | 40 (12.4) | 105 (32.5) | 143 (44.3) | 35 (10.8) |
| *Employment Statuts* | | | | |
| Students | 51 (13.7) | 134 (36.0) | 152 (40.9) | 35 (9.4) |
| Employed | 24 (30.0) | 32 (40.0) | 19 (23.7) | 6 (7.3) |
| Unemployed | 3 (10.3) | 14 (48.3) | 9 (31.1) | 3 (10.3) |

(from 0 = not at all to 4 = extremely). It measures state (STAI-S) and trait (STAI-T) anxiety. High scores indicate high levels of anxiety. For this study, we adopt only STAI-T.

### Alexithymia

The 20-Items Toronto Alexithymia Scale (TAS-20; *Bressi et al., 1996*) was adopted to evaluate alexithymia and its features. Three different dimensions of alexithymia were assessed: difficulty identifying feelings (DIF), difficulty describing feelings (DDF), and externally oriented thinking (EOT). A global score identifies the severity of the alexithymia. A 5-point Likert scale classifies the responses (from 1 = strongly disagree to 5 = strongly agree). Higher scores in the three facets of TAS-20 identify high alexithymic levels.

### Impulsivity

The Barratt Impulsiveness Scale-11 (BIS-11; *Fossati et al., 2001*) is a 30-item questionnaire on a 4-point Likert scale that measures different traits of impulsivity, *i.e.,* attentional impulsivity, motor impulsivity, non-planning impulsivity. Higher scores for each subscale identify higher impulsivity traits.

### Depression

The Beck Depression Inventory (BDI, *Beck et al., 1961*) is a 21-item questionnaire that assesses depression severity, attitudes and symptoms of depression using 4-point Likert scales. Higher scores indicated high depressive symptomatology.

### Social phobia, social anxiety, and social avoidance

The Liebowitz Social Anxiety Scale (LSAS, *Liebowitz & Pharmacopsychiatry, 1987*) is a 24-item questionnaire that assesses Social Anxiety severity, fear, and avoidance of social situations. The scale comprises 11 items related to social interaction and 13 items connected to public performance. Each item requires rating the fear level on a 4-point Likert scale (from 0 = none to 3 = severe), and avoidance in the last week (from 0 = never to 3 =

usually). The LSAS may furnish three measures: social phobia (total score of LSAS), social anxiety, and social avoidance.

## Procedure

An online survey was adopted to collect data from the general Italian population from September to December 2019 (before the spread of the COVID-19 pandemic). All the questionnaires included in the survey are adopted in accordance with published licenses. Before filling out the survey, participants were informed about the general aim of the study, and they had to fill in the informed consent. After a short demographic questionnaire, participants completed the questionnaires. No personal information was collected to guarantee anonymity. All the procedure was approved by the ethical committee of the Department of Dynamic and Clinical Psychology ("Sapienza" University of Rome; protocol number: 0000801) and conformed to the Helsinki Declaration.

## Statistical analysis

Means and standard deviations of continuous variables and frequency and percentage of categorical variables were computed.

Analyses of variances, including age as covariate (ANCOVAs), were performed to investigate the association between groups (No-viewers, No-BW, Moderate BW, Problematic BW) and the psychological variables (trait anxiety, social phobia, social anxiety, social avoidance, depression, alexithymia, impulsivity). Significance was set at $p < 0.05$. The $t$-test for independent samples was used to analyze the differences between groups, and Bonferroni's correction was considered.

Pearson's $r$ correlations among all the psychological variables and BWAQ score were carried out to assess the association between the variables. To suggest a possible predictive model, a multiple linear regression, controlling for age, considered total BW as the dependent variable and psychological domains as predictive factors. Specifically, to avoid redundancy effects in the analyses according to the correlation analysis results, only he global score of the TAS-20 was considered for emotional dysregulation index and total LSAS score for social phobia (including avoidance and anxiety indices). BDI score was considered for depressive symptoms, the STAI score for trait of anxiety, the BIS-11 subscales were included for the different traits of impulsivity.

# RESULTS

## General dimensions

The ANOVA on the age reported a significant difference between the groups ($F_{3,478} = 60.82$; $p = 0.0001$; $\eta p2 = 0.28$). The Problematic BW group was younger than No-BW (mean difference: $-2.87$, $t = -6.25$, $p = 0.0001$) and No-viewers (mean difference: $-4.27$, $t = -8.29$, $p = 0.0001$) groups, but it did not differ from Moderate BW (mean difference: $-0.07$; $t = -0.16$; $p = 0.87$). Moreover, Moderate BW was younger than No-BW (mean difference: $-2.80$; $t = -9.71$; $p = 0.0001$) and No-viewers (mean difference: $-4.19$; $t = -11.33$; $p = 0.0001$). Finally, No-BW was younger than No-viewers (mean difference: $-1.39$; $t = -3.77$; $p = 0.0001$).

**Table 2  Means and standard deviation of psychological characteristics in the four groups of participants.**

|  | Non-Viewers | NO-BW | Moderate BW | Problematic BW | F | p |
|---|---|---|---|---|---|---|
| Age | 24.11 ± 4.76 | 22.71 ± 0.72 | 19.91 ± 2.78 | 19.84 ± 2.81 | 60.82 | 0.0001 |
| Trait anxiety (STAI) | 47.54 ± 10.76 | 45.55 ± 9.51 | 59.93 ± 10.06 | 66.25 ± 8.82 | 76.45 | 0.0001 |
| Social phobia (LSAS total) | 40.09 ± 25.89 | 38.73 ± 23.04 | 65.15 ± 26.91 | 87.45 ± 31.84 | 63.95 | 0.0001 |
| Social anxiety (LSAS subscale) | 23.56 ± 14.51 | 22.58 ± 12.41 | 35.46 ± 13.96 | 44.88 ± 16.04 | 49.22 | 0.0001 |
| Social avoidance (LSAS subscale) | 16.52 ± 12.82 | 16.15 ± 11.45 | 29.70 ± 14.27 | 42.57 ± 17.25 | 67.94 | 0.0001 |
| Depression (BDI) | 10.84 ± 9.85 | 8.64 ± 7.46 | 19.68 ± 11.91 | 34.11 ± 14.27 | 87.31 | 0.0001 |
| Total alexithymia (TAS-20) | 46.72 ± 12.74 | 42.89 ± 12.08 | 59.48 ± 13.02 | 65.02 ± 15.92 | 69.04 | 0.0001 |
| DIF (TAS-20) | 14.73 ± 7.08 | 13.99 ± 6.15 | 21.90 ± 7.19 | 25.41 ± 7.88 | 62.71 | 0.0001 |
| DDF (TAS-20) | 13.65 ± 5.03 | 11.92 ± 4.74 | 16.89 ± 4.76 | 18.20 ± 5.76 | 40.75 | 0.0001 |
| EOT (TAS-20) | 18.33 ± 5.41 | 16.99 ± 4.43 | 20.69 ± 4.84 | 21.41 ± 5.76 | 21.47 | 0.0001 |
| Cognitive impulsivity (BIS-11) | 15.56 ± 3.32 | 15.99 ± 3.35 | 18.52 ± 3.84 | 21.09 ± 3.92 | 38.43 | 0.0001 |
| Motor impulsivity (BIS-11) | 18.52 ± 4.41 | 19.36 ± 4.47 | 21.35 ± 4.40 | 24.57 ± 4.46 | 22.64 | 0.0001 |
| Non-planned impulsivity (BIS-11) | 24.45 ± 5.42 | 24.51 ± 4.64 | 26.31 ± 4.84 | 28.23 ± 5.69 | 9.84 | 0.0001 |

**Notes.**
All the $F$ are reported considering ANOVA without adjustement for age (covariate).

## Mental health dimensions
### Trait anxiety
The ANCOVA on trait anxiety showed significant differences between the groups ($F_{1,477} = 55.06$; $p = 0.0001$; $\eta p2 = 0.26$). Problematic BW reported higher trait anxiety compared to all the other groups (*vs.* Moderate BW: mean difference = 9.30, $t = 5.61$, $p = 0.0001$; *vs.* No-BW: mean difference = 19.97, $t = 11.58$, $p = 0.0001$; *vs.* No-viewers: mean difference = 17.62, $t = 8.67$, $p = 0.0001$). Moreover, Moderate BW reported higher trait anxiety than both No-BW (mean difference: 10.66, $t = 9.38$, $p = 0.0001$) and No-viewers (mean difference = 8.32, $t = 5.53$, $p = 0.0001$). No significant differences emerged between No-BW and No-viewers (mean difference = −2.34, $t = −1.73$; $p = 0.09$) (See Table 2 and Fig. 1).

### Social phobia, social anxiety, and social avoidance
The ANCOVA on the total score of the LSAS questionnaire reported significant between-group differences in social phobia ($F_{1,477} = 46.30$; $p = 0.0001$; $\eta p2 = 0.22$). The Problematic BW group reported higher social phobia than all the other groups (*vs.* Moderate BW: mean difference = 22.26, $t = 5.11$, $p = 0.0001$; *vs.* No-BW: mean difference = 47.42, $t = 10.48$, $p = 0.0001$; *vs* No-viewers: mean difference = 45.43, $t = 8.71$, $p = 0.0001$). Moreover, Moderate BW reported higher social phobia than both No-BW (mean difference = 25.16, $t = 8.43$, $p = 0.0001$) and No-viewers (mean difference = 23.17, $t = 5.86$, $p = 0.0001$). No significant differences emerged in LSAS total score between No-BW and No-viewers (mean difference = −1.98, $t = −0.55$; $p = 0.57$). The same pattern was reported in both LSAS Social Anxiety subscale ($F_{1,477} = 34.19$; $p = 0.0001$; $\eta p2 = 0.17$) and LSAS Social Avoidance subscale ($F_{1,47}7 = 50.98$; $p = 0.0001$; $\eta p2 = 0.24$) (See Table 2 and Fig. 2).
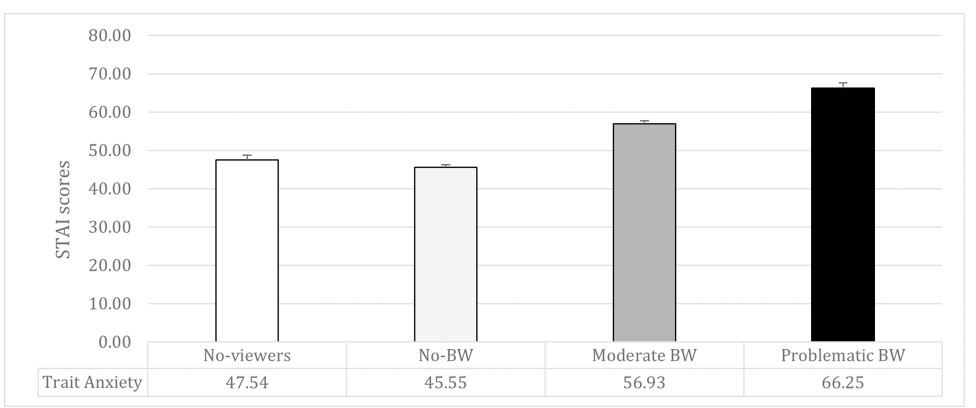

**Figure 1  Means and standard errors of trait-anxiety (STAI scores) in the four groups of participants.**

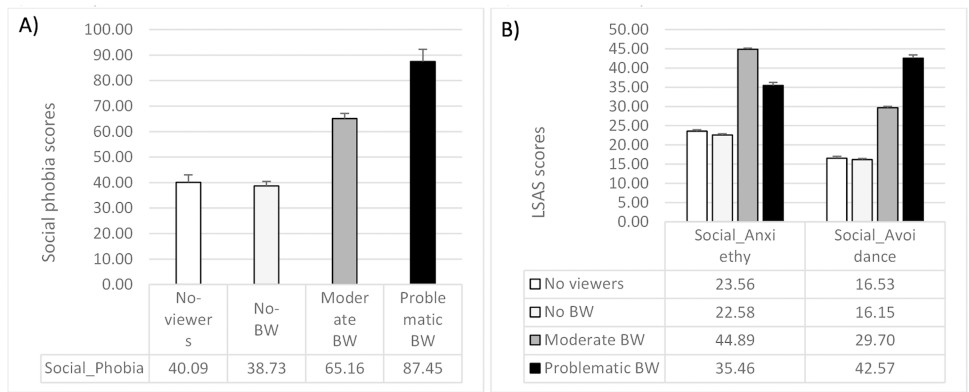

**Figure 2  Means and standard errors of (A) social phobia and (B) social anxiety and social avoidance in the four groups of participants.**

### Depression

The ANCOVA on the global score of the BDI reported a significant between-group difference ($F_{1,477} = 66.17$; $p = 0.0001$; $\eta$p2 $= 0.29$). The Problematic BW group reported higher depression than all the other groups (Moderate BW mean difference $= 14.41$, $t = 8.26$, $p = 0.0001$; No-BW mean difference $= 24.58$, $t = 13.54$, $p = 0.0001$; No-viewers $= 22.09$, $t = 10.56$; $p = 0.0001$). Moreover, Moderate BW reported higher depression than both No-BW (mean difference: 10.16, $t = 8.49$, $p = 0.0001$) and No-viewers (mean difference: 7.68, $t = 4.84$, $p = 0.0001$). No significant differences emerged in the BDI score between No-BW and No-viewers (mean difference: $-2.48$, $t = -1.74$; $p = 0.08$) (See Table 2).

### Alexithymia

The ANCOVA on the global score of the TAS-20 reported a significant between-group difference ($F_{1,477} = 44.66$; $p = 0.0001$; $\eta$p2 $= 0.22$). The Problematic BW group reported higher alexithymia than all the other groups (*vs.* Moderate BW: mean difference $=$

5.49, $t = 2.54$, $p = 0.02$; *vs.* No-BW: mean difference $= 20.43$, $t = 9.09$, $p = 0.0001$; *vs.* No-viewers: mean difference $= 15.78$, $t = 6.09$, $p = 0.0001$). Moderate BW reported higher alexithymia than both No-BW (mean difference $= 14.94$, $t = 10.09$, $p = 0.0001$) and No-viewers (mean difference $= 10.29$, $t = 5.25$, $p = 0.0001$). Finally, No-viewers showed a higher TAS-20 global score than No-BW (mean difference $= -4.64$, $t = -2.63$; $p = 0.02$).

Considering the TAS-20 dimensions, significant differences emerged between the groups in all the subscales (DIF: $F_{1,477} = 38.12$; $p = 0.0001$; $\eta p2 = 0.19$; DDF: $F_{1,477} = 25.16$; $p = 0.0001$; $\eta p2 = 0.14$; EOT: $F_{1,477} = 17.27$; $p = 0.0001$; $\eta p2 = 0.10$). In all three subscales, the Problematic BW showed higher scores, indicating s greater difficulty than No-BW (all the $t > 5.19$; all the $p = 0.0001$) and No-viewers (all the $t > 3.14$; all the $p < 0.0001$). However, while in the DIF, Problematic BW showed greater difficulty than Moderate BW (mean difference $= 2.48$, $t = 2.04$, $p = 0.005$), this difference was not observed in DDF and EOT (all the $t < 1.59$; all the $p > 0.11$). Moderate BW reported higher DIF, DDF, and EOT than No-BW (all the $t > 6.60$; all the $p < 0.0001$) and No-viewers (all the $t > 3.11$; $p = 0.0001$). Finally, higher DDF was reported in No-viewers compared to No-BW (mean difference $= -2.08$, $t = -3.14$, $p = 0.005$), while no differences emerged between these two groups in DIF and EOT (all the $t < 1.98$; all the $p > 0.10$) (See Table 2 and Fig. 3).

### Impulsivity

The ANCOVAs on the BIS-11 subscales reported a significant between-group difference in all three subscales (attentional impulsivity: $F_{1,477} = 23.87$; $p = 0.0001$; $\eta p2 = 0.13$; motor impulsivity: $F_{1,477} = 19.62$; $p = 0.0001$; $\eta p2 = 0.11$; not-planned impulsivity: $F_{1,477} = 5.38$; $p = 0.001$; $\eta p2 = 0.03$).

Problematic BW reported higher attentional and motor impulsivity than all the other groups (all the $t > 4.07$; all the $p < 0.0001$). Moreover, Moderate BW reported higher impulsivity than both No-BW and No-viewers (all the $t > 4.12$; $p = 0.0001$). No significant differences emerged in both the subscales between No-BW and No-viewers (all the $t < 1.45$; $p > 0.14$).

Considering the non-planned impulsivity subscale, the Problematic BW reported higher scores than No-BW (mean difference $= 3.27$, $t = 3.78$, $p = 0.0001$) and No-viewers (mean difference $= 3.11$, $t = 3.12$, $p = 0.01$), but it did not differ from Moderate BW (mean difference $= 1.85$, $t = 2.23$, $p = 0.08$). Moreover, moderate BW showed higher non-planned impulsivity than No-BW (mean difference $= 1.42$, $t = 2.49$, $p = 0.05$). However, no significant differences emerged between Moderate BW and No-viewers (mean difference $= 1.25$, $t = 1.66$, $p = 0.19$) and between No-BW and No-viewers (mean difference $= -0.16$, $t = -0.24$, $p = 0.81$) (See Table 2).

### Pearson's *r* correlation and linear regression model

Pearson's r correlation reported significant association between all the variable assessed (see Table 3). Multiple regression model resulted significant ($R^2$ adjusted $= 0.58$; $F = 69.1$; $p < 0.001$). All the psychological variables, except attentional and non-planned impulsivity, were significantly associated with BW scores (see Table 4).
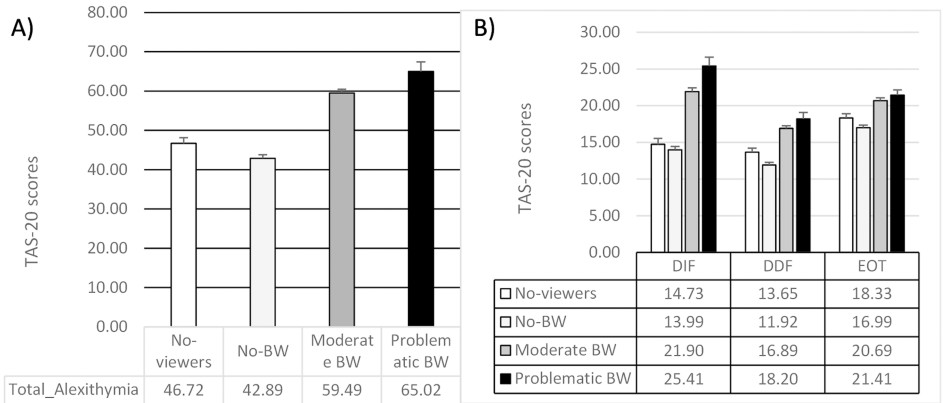

**Figure 3**  Means and standard errors of (A) alexithymia and (B) its facets in the four groups of partici-
pants.

## DISCUSSION

The findings of this study confirmed the association between binge-watching and mental
health, highlighting worse mental health in individuals who presented problematic BW.
The mental health alterations of participants with problematic BW are comparable to that
shared by people with addictive behaviors. Particularly, individuals with moderate and
problematic BW reported higher depression, trait anxiety, social anxiety and avoidance,
emotional dysregulation, and impulsivity. Moreover, respondents with problematic BW
reported significant levels of mental distress, exceeding the threshold values, indicating the
possible presence of a general pathological condition.

Our results suggest that impulsivity, anxiety, alexithymia, social anxiety, and depressive
symptoms were independently associated with problematic binge-watching. Thus, worse
mental health was associated with binge-watching. Our evidence confirmed and extended
previous findings (*e.g.*, *Flayelle et al., 2020*; *Starosta & Izydorczyk, 2020*; *Steins-Loeber et al.,
2020*; *Sun & Chang, 2021*; *Forte et al., 2023*), adding new evidence on the psychological
features associated with the behavioral pattern of BW.

Although previous results are mixed (*e.g.*, some authors found that depression was
associated with a decrease in BW), the positive relationship between negative affect and
problematic BW argues in favor of the hypothesis of BW as a potential maladaptive
emotion-focused coping strategy (*Flayelle et al., 2020*; *Sun & Chang, 2021*). In line with
this hypothesis, binge-watching could be adopted to manage negative emotions, escaping
from reality experienced as hostile (*Panda & Pandey, 2017*; *Sun & Chang, 2021*). Difficulty
in coping adaptively with distressing life events could be expressed by the engagement
in recursive behaviors that could become addictive. In this view, the relationship
between problematic BW and worse mental health, with higher depression and emotional
dysregulation, as well as general and social anxiety and avoidance, seems justified (*Starosta
& Izydorczyk, 2020*; *Vaterlaus et al., 2019*). Association between psychological variables and
problematic behavioral patterns were previously found for other behavioral addictions.

Favieri et al. (2023), *PeerJ*, DOI 10.7717/peerj.15796

**Table 3  Pearson's *r* correlation.**

| | BWAQ Total | TAS-20 Total | DIF | DDF | EOT | STAI | BDI | LSAS Social Phobia | LSAS Social Anxiety | LSAS Social Avoidance | Motor Impulsivity | Non planned Impulsivity |
|---|---|---|---|---|---|---|---|---|---|---|---|---|
| TAS-20 Total | 0.58 | – | | | | | | | | | | |
| DIF | 0.57 | 0.89 | – | | | | | | | | | |
| DDF | 0.50 | 0.86 | 0.70 | – | | | | | | | | |
| EOT | 0.37 | 0.68 | 0.37 | 0.43 | – | | | | | | | |
| STAI | 0.62 | 0.61 | 0.65 | 0.51 | 0.27 | – | | | | | | |
| BDI | 0.62 | 0.59 | 0.63 | 0.47 | 0.29 | 0.81 | – | | | | | |
| LSAS Phobia | 0.58 | 0.56 | 0.56 | 0.51 | 0.27 | 0.67 | 0.62 | – | | | | |
| LSAS Anxiety | 0.54 | 0.56 | 0.57 | 0.50 | 0.24 | 0.63 | 0.60 | 0.96 | – | | | |
| LSAS Avoidance | 0.58 | 0.53 | 0.51 | 0.49 | 0.27 | 0.64 | 0.59 | 0.97 | 0.86 | – | | |
| Attentional Impulsivity | 0.42 | 0.53 | 0.54 | 0.44 | 0.29 | 0.49 | 0.52 | 0.38 | 0.37 | 0.36 | – | |
| Motor Impulsivity | 0.32 | 0.29 | 0.30 | 0.21 | 0.19 | 0.28 | 0.34 | 0.19 | 0.18 | 0.20 | 0.55 | – |
| Non-planned Impulsivity | 0.26 | 0.28 | 0.24 | 0.18 | 0.28 | 0.26 | 0.24 | 0.17 | 0.15 | 0.17 | 0.35 | 0.46 |

**Notes.**
All $p < 0.001$.

**Table 4** Coefficents of the multiple regression model.

| Predictor | t | p |
|---|---|---|
| Alexithymia | 3.06 | 0.002 |
| Trait anxiety | 2.27 | 0.02 |
| Depressive symptoms | 2.67 | 0.008 |
| Social phobia | 4.90 | 0.0001 |
| Attentional impulsivity | −0.47 | 0.64 |
| Motor impulsivity | 2.82 | 0.005 |
| Non-planned impulsivity | 0.27 | 0.79 |

**Notes.**
Results are corrected for age.

For example, it was hypothesized that impulsivity is associated with maladaptive behavior such as binge eating (*Schag et al., 2013*), sex addiction (*Levi et al., 2020*), and gambling disorder (*Hodgins & Holub, 2015*); anxiety and depression are common and cross-sectional to different behavioral addictions (*Karim & Chaudhri, 2012*; *Mastropietro et al., 2022*); alexithymia and emotional dysregulation are reported in behavioral addictions such as Internet and smartphone addiction (*Dalbudak et al., 2013*; *Elkholy, Elhabiby & Ibrahim, 2020*). Further studies should investigate the direction of the relationship between psychological conditions and BW in terms of causality. It is relevant identifying whether the problematic BW impacts mental health, generating this constellation of psychological symptoms, or whether the personality pattern characterized by high depression, anxiety, impulsivity, and emotional dysregulation causes BW as a maladaptive coping strategy.

A second aim of the study was to furnish preliminary support to the I-PACE model in the BW topic. Accordingly, individuals with problematic BW may present a predisposing personality pattern characterized by higher impulsivity, emotional dysregulation (*i.e.,* difficulty in identifying and describing feelings and concrete thought), depression, general anxiety, and social anxiety (with high levels of phobia and avoidance). As preliminary evidence, we confirm a possible predictive role of psychological factors in the variation of BW; specifically, worse psychological conditions increase BW scores, suggesting how worse mental health would represent a risk factor for the problematic expression of BW. Although our study confirms these assumptions, further studies should analyze untreated aspects of the I-PACE, such as cognitive and neurobiological ones.

An interesting insight of this study is related to the adaptive habits of watching TV series as a leisure activity. The behavioral addiction research field is characterized by an increasing number of studies focused on identifying new addictions (*Billieux et al., 2015*). BW has also often been addressed from this perspective (*Forte et al., 2021*). This study helps to understand the vision of TV series without overpathologizing it as well as without minimizing the obvious negative outcomes that can result from a dysfunctional involvement in this activity. Watching TV series could be a leisure and adaptive activity that, if not extreme, could be related to positive mental health and affects general well-being positively (*Granow, Reinecke & Ziegele, 2018*), increasing gratification and satisfaction. Accordingly, the results of this study reported a trend characterized by

a better mental state in individuals who adopted non-problematic BW than individuals who generally do not watch TV series, especially considering emotional dysregulation, depression, and anxiety. Passionate tv series watching could help people feel better and manage their emotions efficiently, indicating a possible role of BW also as an adaptive coping strategy. All the negative consequences associations with poor mental health could emerge when the behavior becomes maladaptive and extreme, expressing traits typical of behavioral addictions (*i.e.,* craving, dependency, anticipation, avoidance), thus supporting a continuum of behavioral pattern from adaptive to maladaptive (*Forte et al., 2021*; *Forte et al., 2023*).

Although the innovative insights of this study are important, some limitations should be highlighted. Firstly, the study design was cross-sectional and retrospective; therefore, causal interpretations cannot be clarified. Consequently, longitudinal studies are needed to assess the causality of the observed associations and the stability. Second, the study was a self-administered, questionnaire–based survey, which could have generated some response bias. In addition, participants were recruited online, and we cannot assume that this resulted in a representative or clinically relevant sample, limiting the generalization of our results. Third, respondents resulted in an unequal gender and age distribution with a higher proportion of females and young participants, and an unbalanced group size, especially considering problematic binge-watchers and no-TV series watchers. Although this distribution is in line with previous studies (*Forte et al., 2021*; *Sun & Chang, 2021*), the results could be less representative of the general population, and further studies should investigate gender and age differences. Finally, for the exploratory nature of the study, a few exclusion criteria were fixed, which might have affected our findings. Future research on the relationship between problematic binge-watching and mental health and its potential mechanisms is requested to overcome these limitations and understand this subject better.

## CONCLUSIONS

Binge-watching remains a new construct that still needs studies. Our results underlined that problematic binge-watching was associated with worse mental health but also highlighted the possible positive role of the BW when adequately adopted. Accordingly, a continuum characterized by a linear trend from the BW as a leisure activity to problematic BW as possible addictive behavior was highlighted. These results confirmed previous suggestions that recommended not considering BW only from a pathological perspective but rather carefully considering the pattern according to other psychological aspects involved in the behavior. To understand the continuum of BW from adaptive to maladaptive behavior, our hypothesis provided that this behavior was characterized by positive mental health when practiced as a leisure/passionate activity. However, it becomes negative when this behavior is exaggerated, characterized by craving and addiction, and interferes with other activities. Understanding this trend is very useful for clinical practice, as it allows us to identify when and on what to intervene without overpathologizing this behavior.

### Funding

This work was supported by the grant "Bandi di Ateneo" of Sapienza University of Rome (number RP11916B79284148). The funders had no role in study design, data collection and analysis, decision to publish, or preparation of the manuscript.

### Grant Disclosures

The following grant information was disclosed by the authors:
"Bandi di Ateneo" of Sapienza University of Rome: RP11916B79284148.

### Competing Interests

The authors declare there are no competing interests.

### Author Contributions

- Francesca Favieri conceived and designed the experiments, performed the experiments, analyzed the data, prepared figures and/or tables, authored or reviewed drafts of the article, and approved the final draft.
- Giuseppe Forte conceived and designed the experiments, performed the experiments, analyzed the data, prepared figures and/or tables, authored or reviewed drafts of the article, and approved the final draft.
- Renata Tambelli conceived and designed the experiments, authored or reviewed drafts of the article, and approved the final draft.
- Manuela Tomai analyzed the data, authored or reviewed drafts of the article, and approved the final draft.
- Maria Casagrande conceived and designed the experiments, authored or reviewed drafts of the article, and approved the final draft.

### Human Ethics

The following information was supplied relating to ethical approvals (*i.e.*, approving body and any reference numbers):

Department of Dynamic and Clinical Psychology ("Sapienza" University of Rome; protocol number: 0000801).

### Data Availability

Raw included results of each participant to the questionnaires adopted in the study, and group classification.

### Supplemental Information

Supplemental information for this article can be found online at http://dx.doi.org/10.7717/peerj.15796#supplemental-information.

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
