# Peer review of "I feel addicted to watching TV series: association between binge-watching and mental health"

_PeerJ, doi:10.7717/peerj.15796_

## Round 0.1 · original submission · Major Revisions

The paper is interesting but needs revisions carefully following reviewers' suggestions.

Reviewer 1 ·

Basic reporting

The problem is presented clearly, and the reference literature appears to be reported consistently: the binge-watching problem is compared to other behavioural addictions. In this vein, the authors collected a series of parameters aimed at describing the phenomenon of binge-watching. However, at a certain point in the discussion, the authors introduce the i-pace model, which sounds to give reason and frame the data.

Experimental design

The protocol capitalised on a series of standardised questionnaires. The data were collected via an online platform guaranteeing the anonymity of the responders.

Validity of the findings

The data analysis appears critical to me: the authors carried out a series of ANOVA for each construct they collected data for. However, the ipace model, which is reported in the conclusion, drove me to think that the questionnaire might be selected differently and that the data might be analysed differently, with a more integrated, model-driven, vision. In fact, no correlation matrix has been proposed, and no result about collinearity is reported.

Additional comments

I would understand why the authors recalled a theoretical model at the end of the work, and not at the beginning. Such a way to recall theory at the end casts a shadow of weakness on the paper.
I would like to see a new analysis of the data in light of the model. I might be wrong, but I suppose that some variables might appear redundant (in fact, no collinearly is reported, or correlational matrix is discussed) or not significant, helping authors to propose a new/adapted model of the phenomena.

·

Basic reporting

The topic is interesting and overall, the work is well-written. However, I suggest the authors check all the typos throughout the text. Here are some I noticed:

- Abstract: after subheadings, according to PeerJ standards, there should be a period (not colons).
- Lines 48-50: a round bracket is missing at the end maybe? In any case, all that text in the brackets might be excessive. Please, consider revising it.
- Line 55: “CAMART”.
- Line 71: comma after “Similarly”.
- Line 141: “questionnaire” should be plural.
- Line 160: “pn2” is ηp2.
- Line 262: round bracket before “Flayelle et al., 2020”
- Figure 3a: “(TAS-20” lacks a round bracket
Line 56-60: this sentence is ambiguous and should be written more clearly.

References: please, remember to provide the references’ DOI if available.

Experimental design

The experimental design and statistical approach are adequate and provide meaningful and robust data that answer the research question. I think it would be also interesting to verify if some of the psychological aspects you studied can predict the problematic BW phenotype. Did you consider running a multiple regression with mental health dimensions as predictors and BW as a response variable?

Material and methods: please, specify the cut-off of BWAQ that discriminate each BW profiles.

Validity of the findings

no comment

---

## Round 0.2 · accepted · Accept

The paper can be accepted for publication.

Reviewer 1 ·

Basic reporting

The problem is presented clearly, and the reference literature appears to be reported consistently: the binge-watching problem is compared to other behavioural addictions.

Experimental design

The experimental design is sound and valid

Validity of the findings

The changes the authors implemented increase the quality of the work.

Additional comments

I think now the paper deserves to be published

·

Basic reporting

No comment

Experimental design

No comment

Validity of the findings

No comment

Additional comments

The authors did a good job and improved the quality of the manuscript. I have no other suggestions.
Congratulations!